
# Agronomic and edaphic drought relations. A semiarid rangeland case.

Juan José Martín Sotoca[1,2], Ernesto Sanz[1,2], Antonio Saa-Requejo[1,3], Rubén Moratiel[1,4], Andrés F. Almeida Ñauñay[1,2], and Ana M. Tarquis[1,2]

[1]CEIGRAM, Universidad Politécnica de Madrid, 28040 Madrid, Spain
[2]Grupo de Sistemas Complejos, Universidad Politécnica de Madrid, 28040 Madrid, Spain
[3] Evaluación de Recursos Naturales, Universidad Politécnica de Madrid, 28040 Madrid, Spain
[4] Grupo AgSystems, Universidad Politécnica de Madrid, 28040 Madrid, Spain

*Correspondence to*: Ana M. Tarquis (anamaria.tarquis@upm.es)

**Abstract.** The dynamic of rangelands results from complex interactions between vegetation, soil, climate, and human activity. In arid and semiarid areas, rainfall coefficients of variability are over 30%. This scenario makes rangeland's condition challenging to monitor, and degradation assessment should be carefully considered to study grazing pressures. In the present work, we study the interaction of vegetation and soil moisture in arid rangelands, using vegetation and soil moisture indexes. We aim to study the feasibility of using water soil moisture (soil drought) as a warning index for vegetation drought. An arid agricultural region in the southeast of Spain, in the province of Almeria (Los Vélez), was selected for this study.

MODIS images, with 250 and 500 m spatial resolution, from 2002 to 2019, were acquired to calculate the anomaly (Z-score) for the Vegetation Condition Index ($Z_{VCI}$) and the Water Condition Index ($Z_{WCI}$). ZVCI was calculated using the Normalised Difference Vegetation Index (NDVI). Soil moisture component (W) was obtained using the Optical Trapezoid Model (OPTRAM). The probability of coincidence of their negative anomalies between $Z_{VCI}$ and $Z_{WCI}$, with lags between them, was calculated. The results show that for specific seasons, the anomaly of the water content index had a strong probability of informing in advance where the negative anomaly of VCI will decrease. Soil water content and vegetation indices show more similar dynamics in the months with lower temperatures (from autumn to spring). In these months, given the low temperatures, precipitation leads the vegetation growth. In the following months, water availability depends on evapotranspiration and vegetation type as the temperature rises and the precipitation falls. The stronger relationship between precipitation and vegetation from autumn to the beginning of spring is reflected in the feasibility of $Z_{WCI}$ to aid the prediction of vegetation index anomalies. During these months, using $Z_{WCI}$ and $Z_{VCI}$ as warning indices are possible for two Spanish semiarid rangeland areas: Los Vélez (Almería) and Bajo Aragón (Teruel). Particularly, November to January showed an average increase of 20-30% in the predictability of vegetation index anomalies. We find other periods of relevant increment in the predictability as March and April for Los Vélez, and July, August and September for Bajo Aragón.



# 1 Introduction

Precipitation and temperature directly influence water balance, causing changes in soil moisture regime which, in turn, influences plant growth. Thus, soil moisture is widely recognised as a critical parameter that links precipitation, temperature, evapotranspiration and NDVI. At the same time, temperature also affects plant phenology and growth directly. Farrar et al.
(1994) studied NDVI, rainfall and model-calculated soil moisture in Botswana. Their results showed that while the correlation between NDVI and precipitation is highest for a multi-month average, NDVI is controlled by soil moisture in the concurrent month. Other research on grassland and woodlands aims to show the link between NDVI and water soil content with different lags (Adegoke & Carleton, 2002; Liu & Kogan, 1996).

The difference between surface soil layers and root zone soil must be noted when studying water soil content. Even though a strong correlation has been shown between these layers (Albergel et al., 2008; Babaeian et al., 2018; Hirschi et al., 2014; Sadeghi et al., 2017). Different responses of NDVI to water soil content are found among vegetation, especially between humid and arid or semiarid areas. This difference is due to the disparities among these areas at root zone soils and surface soil layers (Adegoke and Carleton, 2002; Liu and Kogan, 1996; Wang et al., 2007). NDVI has been shown to have strong links with root zone soil moisture and surface soil moisture in grassland and shrubland in semiarid regions (Guan et al., 2020; Schnur et al., 2010; Wang et al., 2007).

Droughts are often considered into four major types: meteorological, agricultural, hydrological and socioeconomic. Meteorological drought is defined as a reduction of precipitation. Agricultural drought occurs when plants do not have enough available water to meet their requirements; therefore, this drought varies based on the vegetation type. Since this is vegetation specific, some soil water deficits may affect vegetation differently. Typically, there is a lag between soil water deficit and how this is reflected in the vegetation with shorter or wider periods. Hydrological droughts are when the water moving through the ground is significantly reduced. Finally, socioeconomic drought is when a drought affects a community's supply of goods and services. These droughts are sequential in time, increasing the complexity of their impacts and conflicts (Allaby, 2014; American Meteorological Society, 2004, 1997; Wilhite and Buchanan-Smith, 2005).

Remote sensing observation can be used to monitor drought-related variables and assess their effects and impacts from an ecosystem perspective. Precipitation has been studied with several indices (Kim et al., 2009; Mahmoudi et al., 2019), such as the Standardised Precipitation Index (SPI; McKee et al., 1993), Effective Drought Index (EDI; Byun and Wilhite, 1999), or Percent Normal Precipitation Index (PNPI; Willeke et al., 1994). Several indices have been developed to estimate soil moisture (AghaKouchak et al., 2015; Wang and Qu, 2009), such as the Standardised Soil Moisture Index (SSI; Hao and AghaKouchak, 2013), the soil moisture percentile (SMP; Sheffield et al., 2004; Wang and Qu, 2009), and OPTRAM (OPtical TRapezoid Model; Babaeian et al., 2018; Sadeghi et al., 2017). The most common vegetation indices to assess vegetation status is the Normalised Difference Vegetation Index (NDVI), which was selected for this study.

As the drought types are sequential, an alarm index can be developed before more damage is caused. Drought indicators represent different stages of the hydrological cycle, such as precipitation or soil moisture and later impacts can be perceived

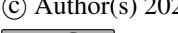



in vegetation water stress. How each stage behaves depends on the particular vegetation or ecosystem. Droughts cannot be
avoided, but their impacts can be reduced by preparing for them. In this respect, different combined indicators present
indices with warning thresholds have been defined (Skees et al., 2001; Wilhite, 2006; Hao and AghaKouchak, 2013;
Sepulcre-Canto et al., 2012; Jimenez-Donaire et al., 2020; Shofiyati et al., 2021). Early warning indices can provide a
drought probability that can be used as a management tool. A proactive approach can be taken in drought risk management
using different risk reduction instruments at different farm or government levels. These instruments include insurance,
irrigation schemes or budget releases. Despite presenting different challenges, early warning systems have already been used
in the past (Canedo Rosso et al., 2018; Desai et al., 2015).

This study's first goal is to understand better the relationships between vegetation indices and water soil content index,
representing the agronomic and edaphic drought in arid rangelands as a complex agricultural dynamical system. The second
goal is to study the feasibility of using the $Z_{WCI}$ as an advanced warning index to predict anomalies in the vegetation activity,
measured by the $Z_{VCI}$. Two semiarid rangelands were selected in Spain, Los Vélez in Almería province and Bajo Aragón in
Teruel province, to accomplish it.

## 2 Material and Methods

### 2.1 Area of study

An extensive area of rangelands was selected in the agricultural region of Los Velez, in Almeria province, southeast of Spain
(Fig. 1). This region presents a mountainous landscape with a slope from 1-14% and with regosols (Xerochrept) soils formed
on limestones and dolomites with poorly developed horizons profile, finding a horizon C immediately above horizon A
(Aguilar Ruiz, 2004).

Los Vélez has an overall Mediterranean climate. It has average monthly temperatures ranging from 5.4℃ to 24.1℃, with an
average yearly precipitation of 373.8 mm. Brushlands Thymus sp. and Rosmarinus sp. coexist with almond trees and cereal
crops (Cueto and Blanca, 1997). The area is characterised by Brachypodietalia phoenicoidis grasses, and its only alliance,
Brachypodion phoenicoidis. These are the more xerophytic and Mediterranean communities of the Festuco-Brometea class.

The Bajo Aragón, in Teruel Province, is in the transitional area between the Iberian System and the Ebro Valley. The
analysed pixels are mainly located in the southern part of the region above 800 m altitude, with complex relief and slopes
between 5 and 30 % (del Palacios Fernandez Montes et al., 2015). It is a scrub landscape with repopulated pine forests and
xeric grasslands (Ibarra et al., 2013). These grasses are dominated by creeping grasses and legumes, which offer a compact
and homogeneous physiognomic appearance bellowing to the alliances described within the order Festuco-Poetalia
lingualatae (San Miguel, 2001). On dry soils, they correspond to Festuco-Brometea class, where Bromus erectus,
Brachypodium phoenicoides, Avenula pratensis, Anthyllis vulneraria and Potentilla neumanniana, among others, are
characteristic.



The soils are cambisols (Calciorthid) and regosols (Xerochrept) formed on limestone and marl, with a developed horizon profile: A-Bw-BC-C (Ibarra Benlloch, 2004) compared to Los Velez. The area analysed has a Mediterranean climate with average monthly temperatures between 0.8 and 29.8 °C and annual rainfall of 648 mm (Grupo de Agroenergética de la ETSI Agrónomos, 2014).

## 2.2 Data Collection

The rangeland pixel selection was provided by Tragsatec in collaboration with Entidad Nacional de Seguros Agrarios (ENESA). They used using Sistema de Información Geográfico de Parcelas Agrícolas (SIGPAC, Fondo Español De Garantia Agraria) and the Mapa Forestal Español (MFE, Spanish Forest Map). Finally, 621 and 1952 pixels defined Los Vélez and Bajo Aragón areas, respectively. NDVI series from 2002 to 2019 for each pixel, with a temporal resolution of 10 days and a spatial resolution of 250 m, were used in this study.

Shortwave infrared reflectance (SWIR, 2105-2155 nm) was collected from MOD09Q1.006 band 7 product from AppEEARS (Team, 2020) to calculate the Shortwave Transformed Reflectance (STR). This product has a 500 m spatial resolution, lower than the one used for the vegetation indices, but a higher spatial resolution was not available for this reflectance band. This band's temporal resolution is 8 days. SWIR series started from 2002 to 2019 to match the time series used in Spanish-indexed agricultural insurance as they provided the selection of the pixels.

Daily meteorological data from the closest meteorological stations (Appendix 1- PONER ALGO MAS?) were also used (Ministerio de Agricultura, 2020; SIAM, 2020). Average temperature and accumulated precipitation were calculated every 10 days to match the NDVI dates.

## 2.3 Estimation of Vegetation and Soil Water Content Indices

NDVI with 10 days temporal resolution and 250 m spatial resolution was used to calculate another vegetation index mainly
used for drought detection, the Vegetation Condition Index (VCI, (Kogan, 1995)). This index was calculated following Eq. [1], where NDVI is each value for each time series, and $NDVI_{min}$ and $NDVI_{max}$ are, respectively, their multiyear minimum and maximum for every ten days:

$$VCI = \frac{\text{NDVI} - NDVI_{min}}{NDVI_{max} - NDVI_{min}} \quad\quad\quad\quad (1)$$

The dynamics of the VCI were analysed and compared to the surface soil moisture index W. To estimate surface soil
moisture, we used the Optimised TRapezoid Model (OPTRAM, (Sadeghi et al., 2017)). This model requires the Shortwave Transformed Reflectance (STR) and the NDVI. STR was calculated using Eq. [2]:

$$STR = \frac{(1 - \text{SWIR})^2}{2 \cdot \text{SWIR}} \quad\quad\quad\quad (2)$$

Firstly, we converted the 8 days-time STR series to 10 days period series like the NDVI and VCI. For every month with 4 values of this time series, every two values were averaged to obtain 3 instead of 4. When a month had 3 values, its values



remained untouched. Secondly, to match the STR and NDVI spatially, every NDVI pixel was given an STR value based on

their centroid proximity. After building a dataset of time series and pixels, this was made for every pixel in all its time series.

We proceeded to calculate the trapezoidal space NDVI-STR. To calculate W, the soil moisture estimator, four parameters are

calculated for each STR-NDVI space $i_d$ and $s_d$ are the intercept and the slope of the dry (upper) edge and $i_w$ and $s_w$ are the

intercept and the slope of the wet (lower) edge (Fig. 2). Using these parameters, NDVI and STR, the W is calculated using

Eq. [3]:

$$W = \frac{i_d + s_d NDVI - STR}{i_d - i_w + (s_d - s_w)NDVI}$$

(3)

After this calculation, all values higher than 1 were replaced by 1, producing a 0-1 range for this index.

We took another step in calculating the Water Condition Index (WCI), submitting the W to the same transformation NDVI

had undergone to calculate VCI. Therefore, we calculated the WCI using Eq. [4]:

$$WCI = \frac{W - W_{min}}{W_{max} - W_{min}}$$

(4)

where W is each value of the time series and $W_{min}$ and $W_{max}$ are, respectively, their multiyear minimum and maximum for

every 10 days.

For both WCI and VCI, anomaly values were calculated using a Z-score as in Eq.s [5]:

$$Z_{WCI} = \frac{WCI - \mu_{WCI}}{\sigma_{WCI}} \quad Z_{VCI} = \frac{VCI - \mu_{VCI}}{\sigma_{VCI}}$$

(5)

where μ is the yearly average and σ is the yearly standard deviation for each year's date. Any trend or seasonal variation is

removed with a Z-score formula, highlighting only the anomaly events.

**2.4 Probabilities of WCI and VCI Anomalies**

The probabilities of passing different thresholds were calculated for these anomalies every 10 days throughout the time

series. Three thresholds (-0.5, -0.7, and -1) were selected following the thresholds used for Standard Precipitation Index

(SPI, McKee et al., 1993) and Standard Precipitation Evaporation Index (SPEI, Vicente-Serrano et al., 2010), indices

commonly used for drought monitoring (Almeida-Ñauñay et al., 2022; Pei et al., 2020). These levels were established based

on previous research for drought identification using the multi-thresholds run theory proposed by He et al. (2016) (Ma et al.,

2022). Firstly, we calculated the probability of a negative VCI anomaly and a negative WCI anomaly for a given time series.

For easier understanding, this will be called base probability. Secondly, the conditional probability going through each level

of the Z_VCI anomaly given a Z_WCI anomaly below -0.3. The conditional probability (understood as frequentist

probability or relative frequency) was calculated using Eq. [6]:

$$P(A|B) = \frac{P(A \cap B)}{P(B)}$$

(6)


Where the probability of A under the condition of B, P(A|B), equals the probability of A and B occurring together divided by
the probability of B.

## 3 Results

### 3.1 Water and Vegetation Condition Index

Firstly, we studied the dynamics of VCI and WCI for Los Velez (Almería province) and Bajo Aragón (Teruel province). Fig.
3A shows the dynamic of the VCI for Loz Vélez (green box plots). The vegetation activity, measured by VCI, increases
from the end of summer until the middle of November, then decreases until the beginning of March and increases again until
the end of May. Therefore, VCI dynamics do not match the seasons: a rise and a drop are observed during autumn, spring
and summer, while VCI dynamics match better during winter. Fig. 3A also shows the behaviour of the soil water content,
measured by WCI (red box plots), during a year. The evolution of both indices is very similar from the end of August to the
beginning of March. However, from March to the end of May, WCI suffers a continuous downfall, unlike the VCI, which
increases in this period.

Fig. 3B shows the dynamic of the VCI for Bajo Aragón (green box plots). In this case, VCI also increases during the end of
summer but extends to the beginning of December, then decreases until the middle of March and increases again until the
end of May. For this area, VCI dynamics do not match the seasons: a rise and a drop are observed during spring and summer,
while VCI dynamics match better autumn and winter. Fig. 3B also shows the dynamic of WCI (red box plots) during a year.
The evolution of both indices is very similar from the middle of August to the middle of March. By contrast, from April to
the end of May, WCI suffers a continuous downfall, unlike the VCI that goes up in this period, with similar behaviour to Los
Vélez.

Following Sanz et al. (2021a), four phases were proposed to analyse the vegetation and soil water content dynamics. For Los
Vélez, VCI and WCI have more similar dynamics in phases 1 and 2. In phase 1, VCI increases due to vegetation activation
after the summer. WCI also rises by the effect of the moisture increment in the soil due to the beginning of rainfalls. The
maximum VCI activity occurs at the end of phase 1 (the middle of November), while WCI peaks during the middle of phase
2 (the end of January). The low temperatures in phase 2 (see Fig. 4A) cause a reduction in plant activity and a continuous
decrease in the VCI graph during this phase. When phase 3 begins, the temperature increases and the WCI graph decreases.
This pattern occurs because the most superficial layers of the soil are losing moisture. In phase 3, precipitations are abundant
(see Fig. 4A), but temperature and evapotranspiration play an essential role in superficial soil water content, obscuring the
relationship between precipitation and water soil content (Sanz et al., 2021a; Wang et al., 2003). The VCI graph has another
peak at the end of phase 3 because of photosynthetic activity growth. In phase 4 both graphs decrease because of high
temperatures and low precipitations in this period of the year.

For Bajo Aragón, VCI and WCI have very similar dynamics in phases 1 and 2 as Los Vélez. In phase 1, VCI and WCI
increase due to vegetation activation and abundant precipitations (see fig. 4B). The maximum VCI activity (the end of phase




1) is delayed for two 10-day periods compared to Los Vélez. In contrast, the maximum WCI at the beginning of phase 2 is advanced four 10-day periods compared to Los Vélez. It should be noted that phase 1 in Bajo Aragón is more prolonged than in Los Vélez. Phases 3 and 4 in Bajo Aragón are very similar to Los Vélez with the same duration and time limits. The only difference is that the VCI maximum in Los Vélez is higher than in Bajo Aragón, probably to the more abundant precipitation

of Los Vélez in phase 2 and the beginning of phase 3 (see fig. 4A).

Secondly, we studied the series of anomaly indices, $Z_{VCI}$ and $Z_{WCI}$, averaged over the two studied areas. In both plots of Fig. 5, it can be observed that $Z_{VCI}$ series are usually smoother than $Z_{WCI}$ series, which present a rougher profile with higher peaks. In fig. 5A, $Z_{VCI}$ serie reflects the long periods of intense drought suffered in Los Vélez the years 2004-2005, 2013-2014 and 2015-2016, when $Z_{VCI}$ reached very low values, approximately -2.0. The drought periods in Bajo Aragón occurred

in 2004, 2011-2012 and 2017 (Fig. 5B).

For Los Vélez (Fig. 6A), the probability of surpassing the first threshold (-0.5) is high in two main periods: the first, during the end of January and the whole of February (with 35%), and the second, during April and the beginning of May (with 40-45%). For the threshold -0.7, the periods of high probability are similar to the -0.5 threshold, although the probability values are lower than those obtained with the other threshold. On the other hand, the probability for $Z_{VCI}$ to surpass below -1.0 is

kept relatively constant most of the year, increasing approximately from 15 % to 25 % from September to October.

For Bajo Aragón (fig. 6B), the probability for $Z_{VCI}$ to surpass the thresholds -0.5 and -0.7 are relatively constant most of the year, although the probability values for the threshold -0.7 are always lower than -0.5, especially in March. The probability for $Z_{VCI}$ to surpass below -1.0 is especially low from December to March when the values are between 10% and 18%.

A similar graphic, but for $Z_{WCI}$ is presented in Fig. 7. In Los Vélez (fig. 7A), the base probabilities for the thresholds -0.5

and -0.7 are more significant than the threshold -1.0, which shows more periods with 0% of base probabilities compared to $Z_{VCI}$. Fig. 7A shows a similar pattern for thresholds -0.5 and -0.7 of $Z_{WCI}$. In both thresholds, the base probabilities are high for two main periods: first, from January to February, and second, in June. For the threshold -0.5, the base probability in these two periods is 35-40%.

In Bajo Aragón (fig. 7B), it is not observed any outstanding period for the probability for $Z_{WCI}$ to surpass the thresholds -0.5

and -0.7. However, for the probability of surpassing the threshold -1.0, we find that March is especially low with values of 5%.

### 3.2 Relationship of VCI and WCI anomalies

For each threshold, the base probability of $Z_{VCI}$ and two conditional probabilities are shown in Figs. 8, 9 and 10. The two conditional probabilities calculated were:

- $P(Z_{VCI} < th \ at \ the \ period \ i \ | \ Z_{WCI} < -0.3 \ at \ the \ same \ period \ i)$:

The probability of an anomaly occurring in VCI at the period i (10-day) under the condition of an anomaly occurring in WCI at the same period. "$th$" are the three different thresholds: -0.5, -0.7 and -1.0. This probability





measures the correlation between anomaly occurrences in the same period without lag. This probability will be named as "lag-0 conditional probability".

-   $P(Z_{VCI} < th$ at the period $i \mid Z_{WCI} < -0.3$ at the period $i - 4$ lags):

The probability of an anomaly occurring in VCI at the period i (10-day) under the condition of an anomaly occurring in WCI 4 10-day periods before. This probability measures the correlation between both anomalies with 4 10-day lag periods. This probability will be named as "lag-4 conditional probability".

In Los Vélez, for all thresholds (Fig. 8A, 9A and 10A), the lag-0 conditional probability is higher from September to April 225 than the base probabilities. Moreover, the lag-4 conditional probability compared to the 0-lag conditional probability is higher from the middle of November to April, except for February for the threshold of -0.5 and December for the threshold of -1.0. The high values in the lag-4 conditional probability are more significant with minor anomalies (-0.5 and -0.7), often reaching from 50% to 65% of probability from November to January (compared to an average of 20-30% of base probability). These probabilities decrease significantly when threshold -1.0 is used, reaching a maximum of 50% in the 230 middle of January. Although the lag-4 conditional probabilities are low for the threshold -1.0, they remain above the base probability.

In Bajo Aragón (fig. 8B, 9B and 10B), the periods where the lag-0 conditional probability is higher than the base probability are not so well-defined as Los Vélez. In fact, unlike Los Velez, we find all the summers with lag-0 conditional probability higher than the base probability for all the thresholds. Regarding lag-4 conditional probability, we find a long period where 235 the one is higher than the base probability. This period corresponds from September to the beginning of February, with probability values reaching 50% from December to January with smaller anomalies (-0.5 and -0.7). For the rest of the year, we can observe some short alternating periods where the lag-4 conditional probability is higher than the base probability, i.e. in June and August for the threshold of -0.5 and June for the threshold of -0.7. As with Los Vélez, the lag-4 conditional probabilities for the threshold -1.0 remain above the base probability.

## 4 Discussion

In Los Vélez, the lag-4 conditional probability is higher than the base probability from October to April for all thresholds. On the contrary, in Bajo Aragón, this period is defined from the middle of July to the beginning of February. Therefore, we find two long periods where the capability of predicting VCI anomalies could be increased in both areas. In Bajo Aragón, this increase in prediction starts before, probably due to more precipitation in June and July than in Los Vélez.

Fig. 11 shows a comparison, for both areas, of the positive difference between the lag-4 conditional probability and the base probability every 10-day of the year and for the three different thresholds (A, B and C). This Fig. summarises the periods where the knowing of $Z_{WCI}$ increases the predictability of $Z_{VCI}$ four lags later compared to the base probability.

For Los Vélez, the increment in predictability from the middle of November to the beginning of February is evident when we consider low anomalies of VCI, i.e. for the thresholds -0.5 and -0.7. The increments in these months reach 25-35% in



November and December. The values range from 10% to 20% during March and April. When we consider high anomalies of VCI, i.e. for the threshold -1.0, the predictability decreases in all the months, with values in the range of 5%-20%. It should be noted that the middle of January keeps high values of predictability, over 30%, similar to the other thresholds.

For Bajo Aragón, we find high positive differences in the same period than Los Vélez, i.e. from the middle of November to the beginning of February for the thresholds -0.5 and -0.7. The increments in these months reach 30-35% in January and the

beginning of February. By contrast, in Bajo Aragón, we find intermittent periods of high predictability values during March (10%) and May (30%). The more outstanding difference between both areas is the additional presence of high predictability during the end of July, August and September. In these periods, we have zero increase in predictability in Los Velez but a 5%-15% increase in Bajo Aragón. These values of positive differences persist in August and September, even in the case of high anomalies of VCI, for the threshold of -1.0.

From October to January, precipitations are more or less abundant in both areas, but they concur with low temperatures. Therefore, this could explain why the predictability of vegetation anomalies using the water soil content index is improved from November to the beginning of February. As precipitation continues to fall, but temperatures begin to rise from February to the end of April, the predictability starts to decrease due to temperature and evapotranspiration play a major importance in water soil content (Cui and Shi, 2010; Sanz et al., 2021a; Wang et al., 2003). For Los Vélez, from May, precipitations

decrease strongly, and temperatures start to increase until the end of summer. In these periods, the water soil content does not give extra information about anomalies in vegetation. By contrast, for Bajo Aragón, we find relatively high precipitations until June and more precipitations than Los Vélez in July and August. This scenario could explain why we have increments in predictability during August and September.

## 5 Conclusions

Soil water content and vegetation indices show more similar dynamics in the months with lower temperatures in both study areas, from autumn to the beginning of spring. In these months, given the low temperatures, precipitation leads the vegetation growth. In the later months, when the temperature rises, soil water availability depends mainly on evapotranspiration and vegetation type.

The stronger relationship between precipitation and vegetation from autumn to the beginning of spring is reflected in the

feasibility of $Z_{WCI}$ to aid the prediction of vegetation anomalies $Z_{VCI}$. During these months, using $Z_{WCI}$ as a warning index is possible for Los Velez and Bajo Aragón. Both are considered semiarid rangelands.

Lag-4 conditional probability measures the probability to occur an anomaly of VCI when an anomaly of WCI happened 4 periods (10-day) before. When we compared this probability with the base probability (the probability to occur an anomaly of VCI), we detected some periods of the year where knowing of an anomaly of water soil content allowed us to increase the

probability of occurrence of an anomaly of vegetation.
The two study areas showed an increase of 15%-35% in the predictability of vegetation index anomalies $Z_{VCI}$ (thresholds -0.5 and -0.7) from November to January. Additionally, in Bajo Aragón, we also find the capability of prediction in August and September due to more precipitation and less temperature in June than in Los Vélez. Therefore, we could increase the predictability of anomalies of VCI in some specific periods if we detect anomalies in the WCI 4 periods before.

This study presents several limitations, some due to remote sensing techniques and others due to the limits of the areas of study. Temporal lengths are limited by satellite data availability, a common problem in remote sensing. Further research is needed to expand to other areas, vegetation, and ecoregions, using more climatic variables to understand vegetation dynamics.

**Data availability**

All data can be provided by the corresponding authors upon request.

**Author contributions**

JJMS, ES and AMT definition of probabilities and thresholds; ES and AFAÑ obtaining series for the WCI and VCI, RM and ASR characterization of zones and bibliographic review, JJMS, ES, AFAÑ calculation programming; all the authors participate in discussion of results; JJMS and AMT review and edited the manuscript

**Competing interests**

The authors declare that they have no conflict of interest.

**Acknowledgements**

The authors acknowledge the support of Clasificación de Pastizales Mediante Métodos Supervisados - SANTO from Universidad Politécnica de Madrid (project number: P220220C024), Caos Hamiltoniano y Sistemas Complejos. Modelos y
Aplicaciones from Ministerio de Ciencia e Innovación (project number: PID2021-122711NB-21).

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




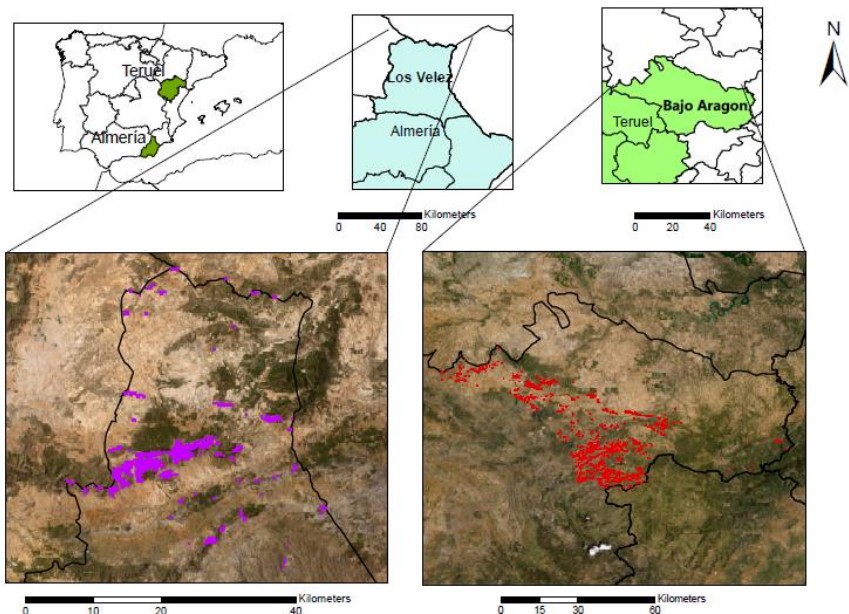

Figure 1: Map representing the selected pixels. In purple, Los Vélez, and in red, Bajo Aragón.



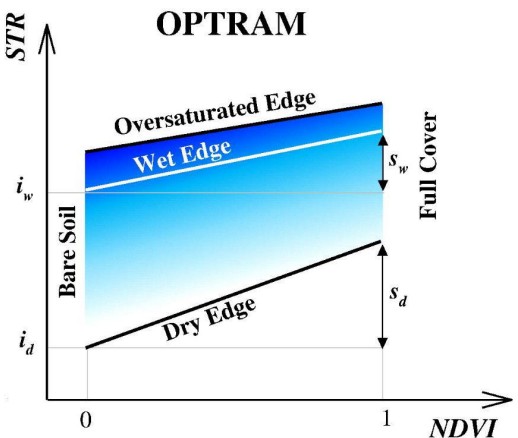

**Figure 2:** Sketch illustrating parameters of the OPTRAM model used in equation 12 to estimate

parameters. Adapted from Sadeghi et al. (2017).
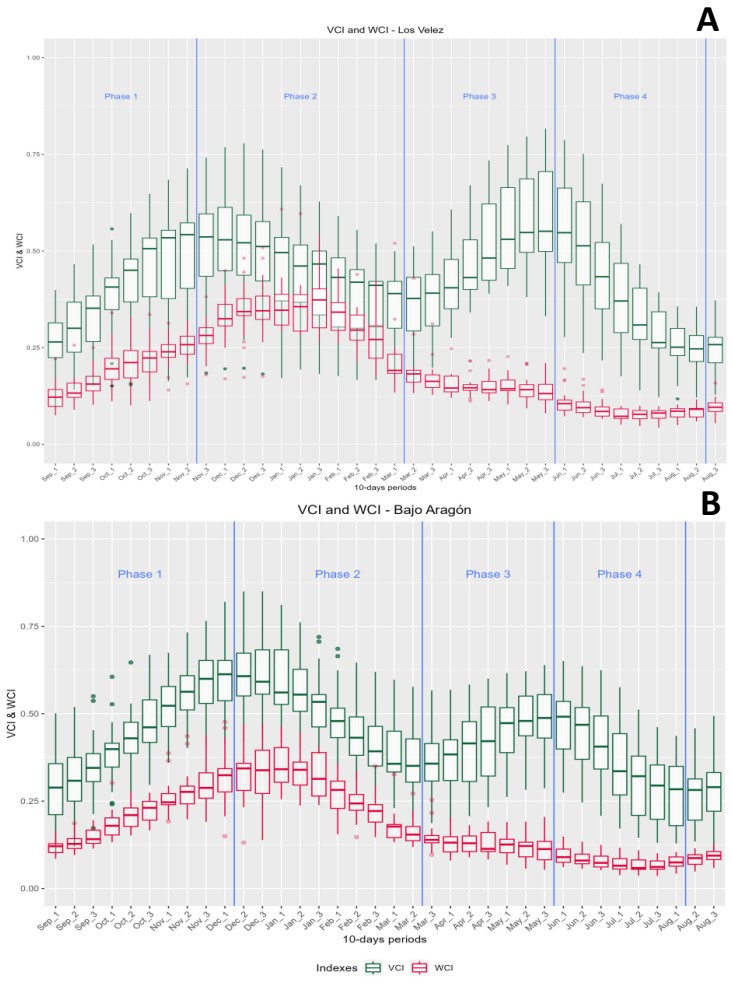

**Figure 3:** Boxplots for vegetation condition index (VCI) and water condition index (WCI) for Los Vélez (A) and Bajo Aragón (B) every 10 days of the year. The blue vertical lines represent the phases split based on the VCI dynamics.


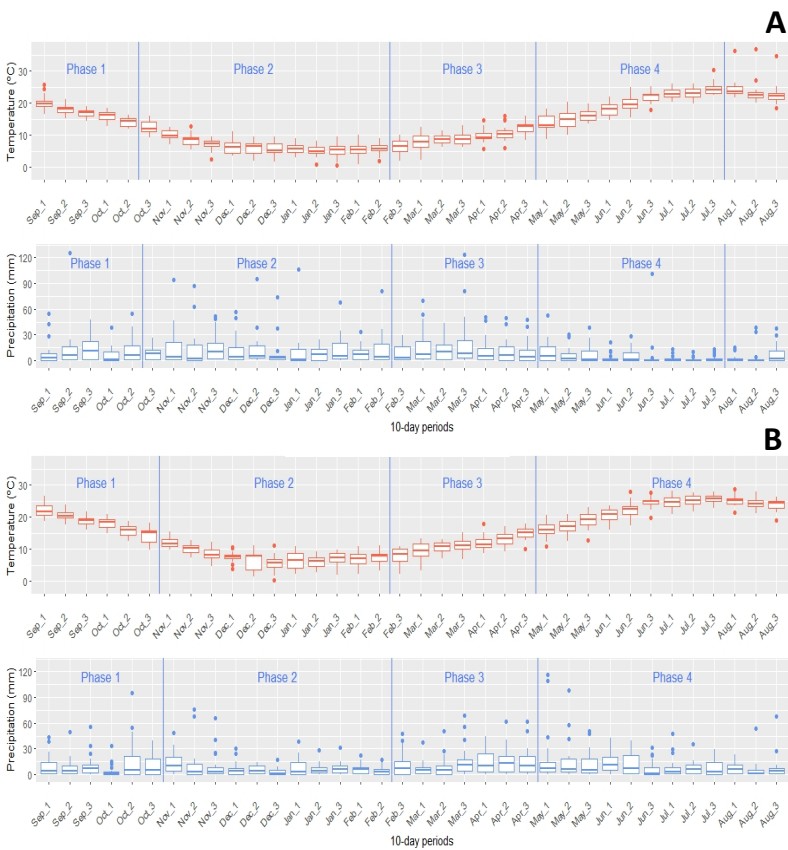

**Figure 4**: Boxplots for temperature (orange) and precipitation (blue) for Los Vélez (A) and Bajo Aragón (B) every 10 days of the year. The blue vertical lines represent the phases split based on the VCI dynamics.

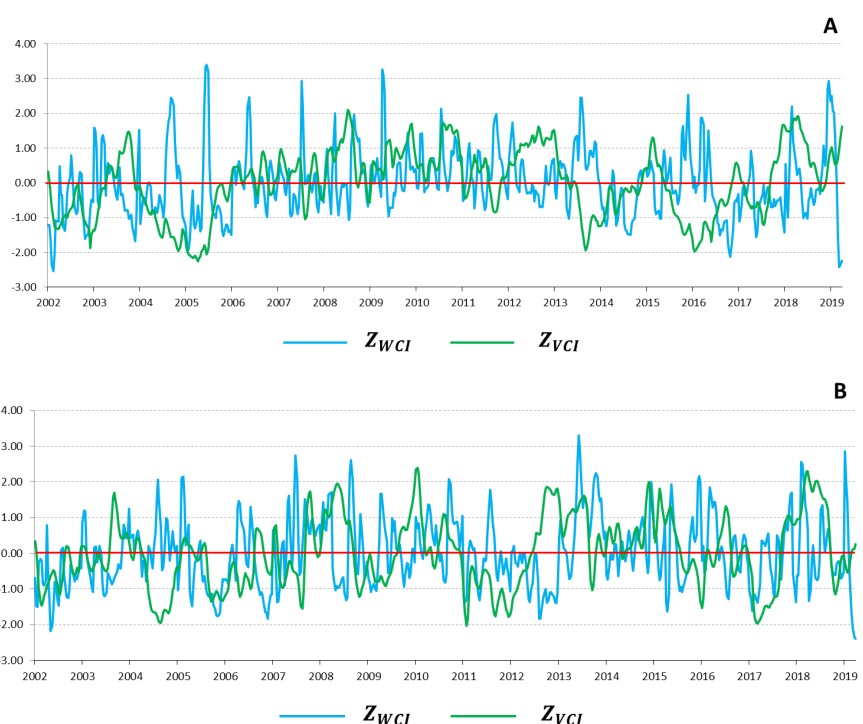

**Figure 5**: Z score series ($Z_{VCI}$ and $Z_{WCI}$) for the average of the selected pixels of Los Velez (A)
and Bajo Aragón (B).

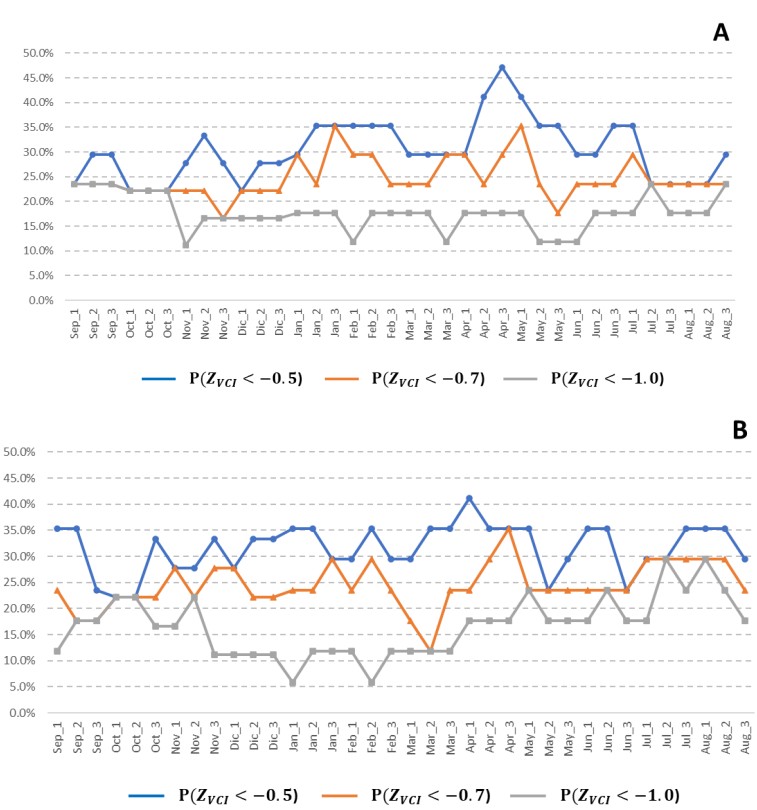

**Figure 6:** Base probabilities for the $Z_{VCI}$ to pass below the three thresholds every 10-day for
Los Velez (A) and Bajo-Aragón (B). Threshold -0.5 in blue, -0.7 in orange, and -1.0 in grey.

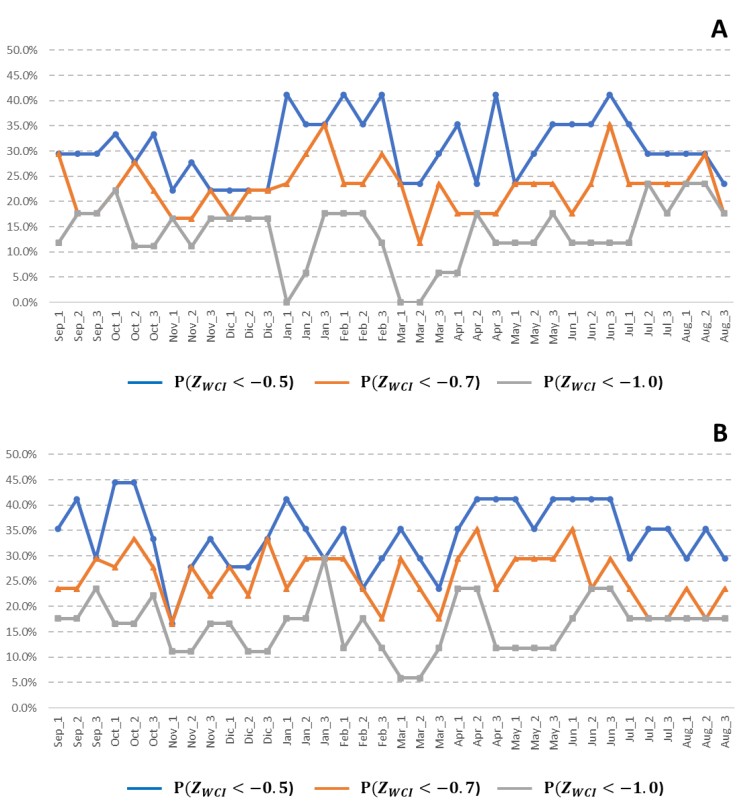

**Figure 7:** Base probabilities for the $Z_{WCI}$ to pass below the three thresholds for each 10-day period for Los Velez (A) and Bajo-Aragón (B). Threshold -0.5 in blue, -0.7 in orange, and -1.0 in grey.

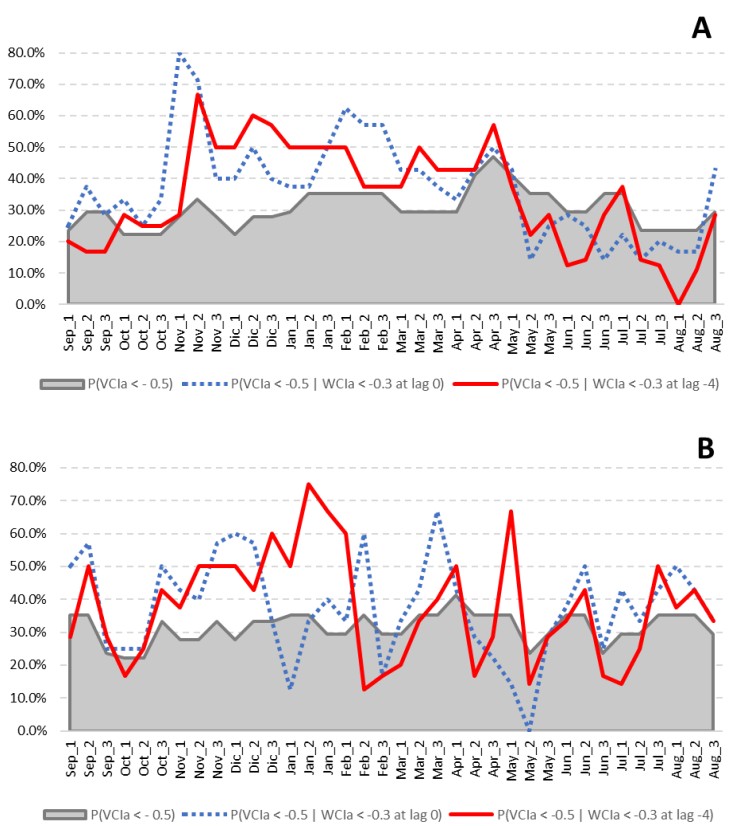

**Figure 8:** Base probability for $Z_{VCI}$ to be below -0.5 (grey), conditional probability without lag (dotted blue) and condition probability for lag -4 (red) in Los Velez (A) and Bajo-Aragón (B).

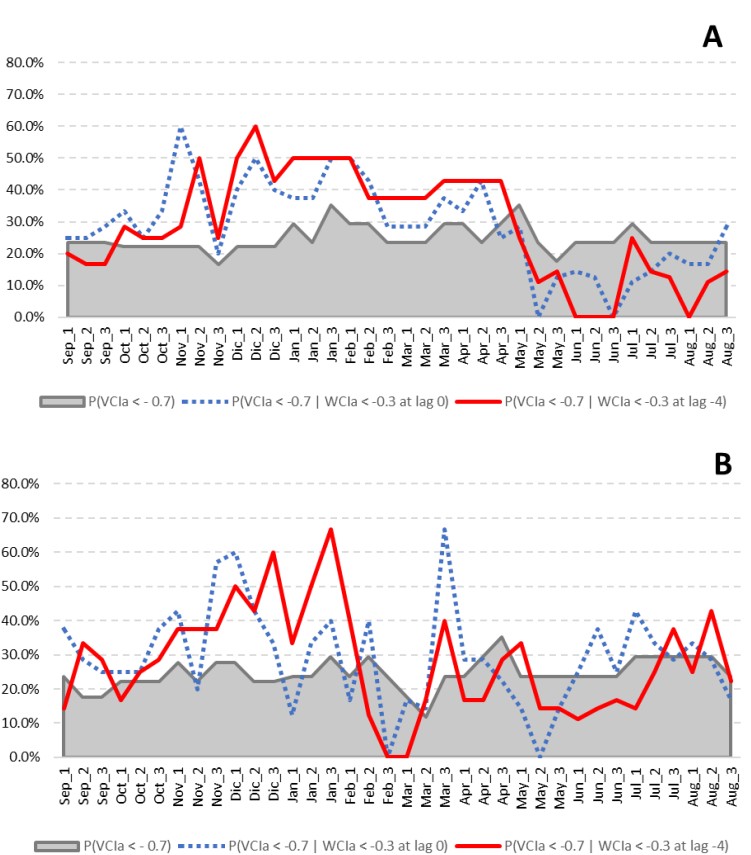

**Figure 9:** Base probability for $Z_{VCI}$ to be below -0.7 (grey), conditional probability without lag (dotted blue) and conditional probability for lag -4 (red) in Los Velez (A) and Bajo-Aragón (B).


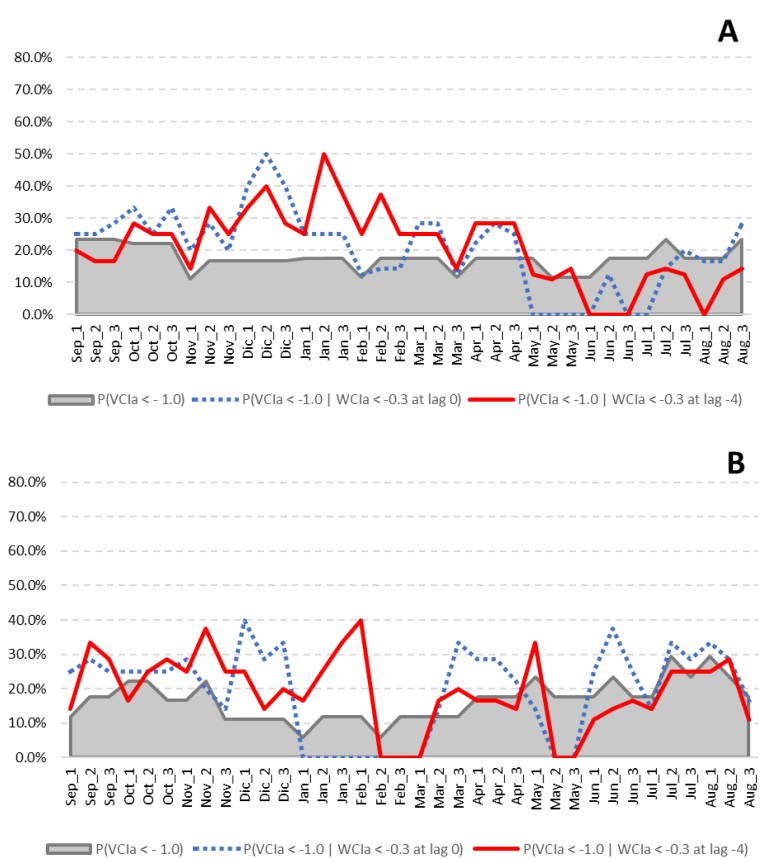

**Figure 10:** Base probability for $Z_{VCI}$ to be below -1.0 (grey), conditional probability without lag (dotted blue) and conditional probability for lag -4 (red) in Los Velez (A) and Bajo-Aragón (B).

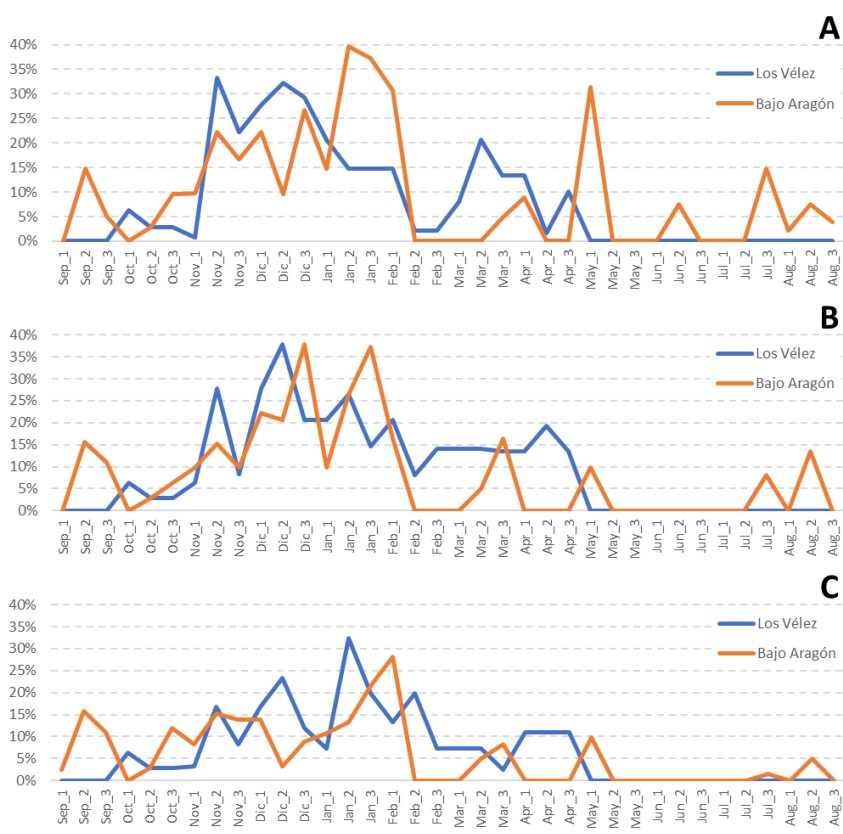

**Figure 11:** Comparison of positive differences (lag-4 conditional probability minus base

probability) for both areas of study and the thresholds -0.5 (A), -0.7 (B), and -1.0 (C).