# Peer review of "Agronomic and edaphic drought relations. A semiarid rangeland case."

_Natural Hazards and Earth System Sciences, 2023_

## Author Comment (AC1)

**REFEREE 1**

**Authors:** We are very grateful to the referee for the comments and suggestions to improve the study.

**Specific Comments**

Ln 14. Can you explain why you are using the term "water soil moisture" in the following phrase "We aim to study the feasibility of using water soil moisture (soil drought)"

**Authors:** There is a mistake in the phrase. It says:

"We aim to study the feasibility of using soil moisture anomalies as a warning index for vegetation or agricultural drought."

Ln 18. Please choose between using ZVCI or ZVCI. You should keep one of those and don't change through the manuscript.

**Authors:** We have corrected these typos throughout the manuscript. Apologise.

Ln 22. VCI = ZVCI? Please be consistent with acronyms throughout the manuscript.

**Authors:** We are talking about VCI anomalies. VCI anomalies are calculated using the VCI's Z-score.

VCI anomalies = Z-score of VCI = $Z_{VCI}$

**English comments**

Ln 24. Precipitation leads TO vegetation growth. OR precipitation grows vegetation.

**Authors:** Thank you very much. We have corrected the mistake. Now it says: "In these months, given the low temperatures, precipitation leads to vegetation growth."

Ln 37. Please use "soil water content". Please modify all the rest by yourself.

**Authors:** Thank you very much. We have changed the concept "soil water content" by "soil moisture content" because it is more adequate in this study.

Ln 46. I think a reference is needed for the types of droughts.

**Authors:** We have arranged the introduction. Now you can find it in Ln 32.

"Drought is the principal climatic hazard to arid and semiarid Mediterranean grasslands and causes physical suffering, economic losses, and environmental degradation. Droughts are often divided into four major types: meteorological, agricultural, hydrological, and socioeconomic

(Allaby, 2014; American Meteorological Society, 2004, 1997; Wilhite and Buchanan-Smith, 2005). Meteorological drought is defined as an accumulated departure of precipitation from normal or expected. Agricultural drought is related to soil moisture availability affecting vegetation growth. Precipitation and temperature directly influence water balance, causing changes in soil moisture regime, which, in turn, influences plant growth. Thus, soil moisture is widely recognised as a critical parameter that links precipitation, temperature, evapotranspiration, and vegetation status. At the same time, temperature also affects plant phenology and growth directly. Therefore, agricultural drought constitutes a combination of climate, soil, and vegetation factors.

Hydrological droughts occur when the water moving through the ground is significantly reduced. Finally, socioeconomic droughts occur when a drought affects a community's supply of goods and services. These droughts are sequential in time, increasing the complexity of their impacts and conflicts."

We are open to any suggestion to add more adequate references.

Ln 73. In line 46 you mention four types of droughts. Now, you are using a "new term," edaphic drought. Why is it not mentioned before?.

**Authors:** We have eliminated the term edaphic drought in all the manuscript (including the title) and only keep agricultural or vegetation drought defined in the introduction.

The aim of this study is not to introduce a new concept (edaphic drought) but to study the relationships between vegetation and soil moisture anomaly indices.

Ln 101. They use using?. Please correct.

**Authors:** Thank you very much. We have corrected the mistake.

Ln 100. It is not clear why they choose those pixels.

**Authors:** Tragsatec company, in collaboration with Entidad Nacional de Seguros Agrarios (ENESA), depending on the Agricultural Ministry of Spain, selected these pixels as semiarid rangelands to be used in the context of the National Rangeland Insurance. Firstly, pixels categorised as rangeland were selected using the SIGPAC. From this previous selection, pixels with a tree coverage higher than 15% were discarded to ensure a low tree coverage based on the MFE (Spanish Forest Map).

Ln 110. (Appendix 1- PONER ALGO MAS?). This is not the first one, I think this research paper needs to be resubmitted. There are too many typos; see above.

**Authors:** Thank you very much. We have corrected the mistake.

Ln 115. You defined the vegetation condition index with ZVCI in the abstract and have now changed it to VCI.

**Authors:** In the abstract, we were talking about VCI anomalies measured by the Z-score of VCI, i.e. $Z_{VCI}$. We have improved these definitions to clarify the concepts better.

Ln 118. Eq 1 is dependent on NDVI and Eq. 3 is dependent on NDVI. Do you think this is tricky if you are looking for a correlation between them (WCI and VCI)?

**Authors:** We have tried to clarify this point with the following paragraph added to the text:

"Note that W depends on STR and ultimately from SWIR. NDVI is only a scale factor to determine $STR_{min_i}$ and $STR_{max_i}$. Even there is a linear relationship between STR and W (Eq. 3), this is not found between NDVI and W, as the dry and wet edges are not parallel lines (Fig. 2). A time-lagged cross-correlation analysis was performed to check that W estimation does not impose a significant relation with NDVI that could bias the estimation of the probabilities applied in this work."

Ln 132. What does the soil water content of one mean?. Is it possible to have soil water content of 0?

**Authors:** The formula of W is: $W_i = \frac{STR_i - STR_{min_i}}{STR_{max_i} - STR_{min_i}}$

Therefore W = 1 means that $STR_i = STR_{max_i}$, the maximum moisture contents in the studied soils (wet edge). By contrast, W = 0 means that $STR_i = STR_{min_i}$, the minimum moisture contents in the studied soils (dry edge).

Ln 144. Which anomalies? I think it is explained below. Please correct the entire paragraph.

**Authors:** We have corrected the paragraph. Now it says:

"The probabilities of surpassing different thresholds were calculated for WCI anomalies ($Z_{WCI}$) and VCI anomalies ($Z_{VCI}$) every 10 days throughout the time series."

Ln. 151 See acronyms.

**Authors:** We have reviewed all the section 2.4.

Ln 153. Instead of A or B, you should use the acronyms presented before.

**Authors:** We have reviewed all the section 2.4.

Final comments.

Why do you not use a method to evaluate the precision of W computation? Measuring surface soil water content is not difficult.

**Authors:** The aim of this study is to show relationships between vegetation and soil moisture based on accepted remote sensing indices such as NDVI and W, and their condition indices, VCI and WCI, respectively.

Section 2.4 is not well written and is essential for the manuscript.

**Authors:** We have reviewed all the section 2.4.

The methodology does not specify how to verify the results. The methodology is only based on indexes previously published; where is the novelty? On the conditional probability?

**Authors:** In this study, we try to show a relationship between vegetation and soil moisture anomaly indices depending on months. This can be used to increase the probability of predicting an anomaly in vegetation indices (agricultural drought) and work as an earlier warming index. Though conditional probability is a classic in statistical terms, we have only found a close study to our work that we have mentioned in the introduction: Hao et al. (2021).

---

## Author Comment (AC2)

**REFEREE 2**

**Authors:** We are very grateful to the referee for the comments and suggestions to improve the study.

General comments:

The paper intended to analyze the relationship between agronomic and edaphic drought in a semiarid rangeland area. The use of several remote sensing indexes is the primary source of data to provide this analysis. The paper's general idea is valuable but not novel as vast literature uses this kind of tool to analyze such relationships at different scales and under different biomes. The main novelty of the paper is perhaps not fully addressed in the introduction.

The causal mechanisms behind the soil and vegetation drought expression are not fully addressed, probably because of the limitation of the pixel resolution that restricts the specific vegetation identifications and the lack of soil hydraulic properties or soil moisture spatial information. No ground data is used, severely limiting the robustness of the remote sensing analysis. Maybe the general idea and the title of the manuscript should be more related to comparing the dynamics of physically different remote sensing drought indexes to detect agricultural drought, which is more related to objective 2. If so, the study's outreach is limited but valuable, maybe for a letter paper.

**Authors:** We have eliminated the term edaphic drought in all the paper (including the title) and only keep agricultural or vegetation drought defined in the introduction.

The aim of this study is not to introduce a new concept (edaphic drought) but to show relationships between vegetation and soil moisture based on accepted remote sensing indices such as NDVI and W. In addition to this we try to show there is a relationship depending on seasons between vegetation and soil moisture anomaly indices and this can be used to increase the probability to predict an anomaly in vegetation indices (agricultural drought).

In addition, the documents presented several grammar, format, and presentation flaws that made it difficult to follow some essential ideas.

**Authors:** We have reviewed all the sections to improve the paper.

All these reasons support the decision to reject this paper in its current form. However, I put some specific comments to enhance the document for a possible resubmission.

**Authors:** Thank you very much for your comments. We hope that the present version of this manuscript will reach the scientific level for this special issue.

Abstract: The justification of the problem is not well accomplished; why does the 30% precipitation variability represent a challenge?

**Authors:** High precipitation variability provokes vegetation water stress. The aim of this study is to find relationships between moisture (water) stress and vegetation stress throughout remote sensing anomaly indices. Beside it, we have changed that phrase for the following:

"**Abstract.** The dynamic of rangelands results from complex interactions between vegetation, soil, climate, and human activity. In arid and semiarid areas, temporal variation of precipitation significantly influences this ecosystem."

What is the relation between degradation and rainfall variability?

**Authors:** The lack of water in certain periods and between different years causes a limitation to the vegetation cover and in addition, when storms occur, great erosion is generated in the poorly protected soil.

Introduction:

References are too old; I suggest keeping only references for the last 10 years. At least, it was a singular paper. The manuscript requires an extensive English revision before being resubmitted, and it is hard to follow the ideas in this current form. I suggest including a more comprehensive explanation and justification of how agronomic and edaphic drought relations can be propagated. The literature remains poor; please add more relevant and actual papers on the subject. Also, an explanation of what details about the OPTRAM index should be included. Why could this index detect something different from the NDVI?

**Authors:** We have eliminated some references and included some new ones. We have eliminated the concept "edaphic drought" because it is not the objective of this study. We have also included a detailed explanation of OPTRAM methodology to build the W index.

It is unclear and not correctly justified why using an arid rangeland could be singular to study these relationships.

**Authors:** In this study we deal with semiarid rangelands but in future research we will deal with other kind of rangelands (arid, humid, etc). On the other hand, semiarid and arid rangelands are more susceptible to changes in their precipitation and humidity regimes, especially given their commonly degraded status.

The definition of edaphic drought is missing.

**Authors:** We have eliminated the concept "edaphic drought" because it is not the objective of this study.

Methods:

The rationale for choosing the different remote sensing indexes is completely missing. You can use a diagram to support the text to better explain your procedure's logic.

**Authors:** We have modified the abstract and the Introduction to clarify this point.

Scientific names should be put in italics.

**Authors:** We have put scientific names in italics.

A description of root depths and the hydraulic properties of soils should be convenient.

**Authors:** In the section "Area of Study" we have included a brief description of the soil types in the studies areas.

The climate description is poor; please use an international reference based on the Koeppen classification.

**Authors:** In the section "Area of Study" we have included a brief climate description in the studies areas.

Now it says: "Los Vélez has an overall Mediterranean climate (Bsk according to Köppen classification). It has average monthly temperatures ranging from 5.4°C to 22.7°C, and average yearly precipitations between 330 and 390 mm (Grupo de Agroenergética de la E.T.S.I. Agrónomos, 2014a)… Bajo Aragón has a Mediterranean climate (Bsk to Csa according to Köppen classification) with average monthly temperatures between 0.8°C and 29.8°C and annual rainfall of 648 mm."

The explanation of pixel selection is very unclear; please explain how you merge the information better.

**Authors:** Tragsatec company in collaboration with Entidad Nacional de Seguros Agrarios (ENESA) depending on the Agricultural Ministry of Spain selected these pixels as semiarid rangelands to be used in the context of the National Rangeland Insurance. Firstly, pixels categorized as rangeland were selected using the SIGPAC and from this previous selection, pixels with a tree coverage higher than 15% were discarded to ensure a low tree coverage, based on the MFE (Spanish Forest Map).

You must put the complete names of the MODIS products used in the study, for instance the NDVI product. How did you deal with clouds? Do you use the Quality Assessment data?

**Authors:** We have included the names of the MODIS products. Now it says: "The NDVI was extracted from the product MOD09GQ of the TERRA satellite and MYD09GQ of the AQUA satellite, which provide the reflectance of the red and near-infrared bands with which the index was built… Shortwave infrared reflectance (SWIR, 2105-2155 nm) was collected from MOD09Q1.006 band 7 product from AppEEARS (Team, 2020) to calculate the Shortwave Transformed Reflectance (STR)."

Do you resample the NDVI to the STR resolution or vice versa? please clarify.

**Authors:** Section 2.2 Data Collection says: "This product has a 500 m spatial resolution, lower than the one used for the vegetation indices, but a higher spatial resolution was not available for this reflectance band."

Section 2.3 Estimation of Vegetation and Soil Moisture Content Indices says: "Secondly, to match the STR and NDVI spatially, every NDVI pixel was given an STR value based on their centroid proximity."

Furthermore, we have included the following phrase to better explain this point: "Therefore, STR spatial resolution (500 m) was adapted to NDVI spatial resolution (250 m)."

A better map of the study area is necessary to understand the context better, please improve the one chosen.

**Authors:** We have improved the map (Fig. 1) including different colours to identify every studied area. Now light blue corresponds to Los Vélez (A) and green corresponds to Bajo Aragón (B) in the map of Spain.

We do not know if the referee would consider incorporating a context map, for example of the western Mediterranean like the one we put here below. We are open to suggestions.

[Figure]

A climograph could be added, for example, and the land cover present in the selected area.

**Authors:** Climograms can be included, but they would reflect the precipitation and temperature information in another format. This information is already included on 10-days series in the work. The type of climate is already included by the Köppen classification.

We have included here the climograms. If the referee believes that should be incorporated we will.

[Figure]

**Authors:** The land cover present in the selected areas has been included in the manuscript: Now it says for Los Vélez: "In this area, 47% of the surface is dedicated to crops, 31% is forestry and 26% is scrub, pastures and meadows.".

And for Bajo Aragón: "In this area, 41% of the surface is dedicated to crops, 21% is forestry and 38% is scrub, pastures and meadows.".

Results:

The results are attractive and are the main strength of the paper, however, they are difficult to connect to the main objectives. Maybe you could enhance the idea of early detection of a hazard.

**Authors:** Thank you very much. The idea of early detection and prediction increment is developed in the Discussion section.

The Discussion sections reflect the poor level of analysis displayed in the document, with only three references to confront the results.

**Authors:** We have included this new paragraph with two new references:

"Early warning system are being developed for famine, agricultural yield, and drought. Recently remote sensing has been used in this regard and is more commonly used for rangeland monitoring. In this paper we merge remote sensing monitoring with early warning indices to inform managers and rangers (Rembold et al., 2019; Haigh et al., 2019)."

Specifics comments:

Line 11: To communicate the idea of precipitation variability, I suggest using another metric or explaining better what you mean by 30% of the coefficient of variability.

**Authors:** Now it says: "In arid and semiarid areas, temporal variation of precipitation significantly influences this ecosystem."

Line 13: what is water soil moisture? Please use a widely used terminology.

**Authors:** There is a mistake in the phrase. Now it says: "We aim to study the feasibility of using soil moisture anomalies as a warning index for vegetation or agricultural drought."

Line 17: Please explain why you selected two MODIS resolutions.

**Authors:** This is explained in Section 2.2 Data Collection, as shown below.

"Shortwave infrared reflectance (SWIR, 2105-2155 nm) was collected from MOD09Q1.006 band 7 product from AppEEARS (Team, 2020) to calculate the Shortwave Transformed Reflectance (STR). This product has a 500 m spatial resolution, lower than the one used for the vegetation indices, but a higher spatial resolution was not available for this reflectance band."

Line 21. What specific season?

**Authors:** We have changed "season" by "months". At the end of the abstract says: "The stronger relationship between vegetation and precipitation from autumn to the beginning of spring is reflected in the feasibility of $Z_{WCI}$ to aid the prediction of $Z_{VCI}$. During these months, using $Z_{WCI}$ and $Z_{VCI}$ as warning indices are possible for both areas studied. Notably, November to the beginning of February showed an average increase of 20-30% in the predictability of vegetation anomalies knowing moisture soil anomalies 4 lags (period of 10 days) later. We found other periods of relevant increment in the predictability, such as March and April for Los Vélez, and from July to September for Bajo Aragón."

Line 34, I suggest explaining that plant growth can be related to NDVI. The Normalized Difference Vegetation Index must be described with its entire name.

**Authors:** We have reorganised the introduction:

Now it says Ln 44: "Remote sensing observation has been increasingly used to monitor drought-related variables and assess their effects and impacts from an ecosystem perspective (Liu & Kogan, 1996). Precipitation has been studied with several indices (Kim et al., 2009; Mahmoudi et al., 2019), such as the Standardised Precipitation Index (SPI; McKee et al., 1993), Effective Drought Index (EDI; Byun and Wilhite, 1999), or Percent Normal Precipitation Index (PNPI; Willeke et al., 1994). Among the vegetation indices, Normalised Difference Vegetation Index (NDVI) is the one most often used for monitoring environmental conditions (e.g. grassland status, land degradation, desertification, and drought) all over the world (Hassan et al., 2018). Kogan (1995) proposed a vegetation condition index (VCI) that utilised historical maximum and

minimum NDVI data to measure vegetation conditions against the historical worst situation. This VCI has been successfully used for drought monitoring (Baniya et al., 2019 and references therein)."

Line 35 Farrat et al., 1994 studied the relationship between NDVI, rainfall, and soil moisture. This is a very old reference, please provide more actual references.

**Authors:** We have eliminated this reference and included two more actual:

Sharma, M., Bangotra, P., Gautam, A.S. et al. Sensitivity of normalized difference vegetation index (NDVI) to land surface temperature, soil moisture and precipitation over district Gautam Buddh Nagar, UP, India. Stoch Environ Res Risk Assess 36, 1779–1789 (2022). https://doi.org/10.1007/s00477-021-02066-1, 2022.

Felegari, S., Sharifi, A., Moravej, K., Golchin, A., Tariq, A. Investigation of the relationship between ndvi index, soil moisture, and precipitation data using satellite images. In book: Sustainable Agriculture Systems and Technologies, Wiley Online Library, pp.314-325, 2022. https://doi.org/10.1002/9781119808565.ch15, 2022.

**Authors:** Now, in Introduction, Ln 67: "Adegoke & Carleton (2002) aimed to show the link between NDVI and water soil content with different lags. This study obtained stronger relations with soil moisture data that lagged to the vegetation indices by up to 8 weeks. This result implies that soil moisture may be a valuable predictor of warm-season satellite-derived vegetation conditions. Recently, Felegari et al. (2022) showed that plant indices such as NDVI have a delayed response to soil moisture. In most cases, soil moisture data and other meteorological characteristics strongly correlate with these indices in a short period. In a related study, Sharma et al. (2022) examined the trends in MODIS/TERRA derived NDVI and its correlation with Land Surface Temperature (LST), Soil Moisture (SM), and precipitation over Gautam Buddh Nagar (India) during the period 2005–2018. The correlation between NDVI and LST was higher than that of NDVI with SM and precipitation."

Line 37 what are the main findings of the papers you referred to?

**Authors:** Now, it says: ""Adegoke & Carleton (2002) aimed to show the link between NDVI and water soil content with different lags. In this study, stronger relations were obtained with soil moisture data that are lagged by up to 8 weeks with respect to the vegetation indices, implying that soil moisture may be a useful predictor of warm season satellite-derived vegetation conditions".

Line 39 and 40: is very difficult to follow the idea, strong correlation of what? The soil moisture between soil surface and lower layers?

**Authors:** We understand that in a daily context and extreme meteorological situations the relationship between soil moisture satellite indices and root-zone soil moisture will probably not occur. But in a decennial-monthly-seasonal context it does happen. We have included in Introduction Ln 60:

"The difference between surface soil layers and root zone soil must be noted when studying water soil content (Hirschi et al., 2014). Even though a strong correlation has been shown between these layers (Albergel et al., 2008; Babaeian et al., 2018). Different responses of NDVI to water soil content are found among distinct vegetation types, especially between humid and arid or semiarid areas. These differences are due to the disparities among these areas at root zone soils and surface soil layers (Adegoke and Carleton, 2002; Liu and Kogan, 1996; Wang et al., 2007). NDVI has been shown to have strong links with root zone soil moisture and surface soil moisture in grassland and shrubland in semiarid regions (Guan et al., 2020; Schnur et al., 2010; Wang et al., 2007)."

For example, Albergel et al (2008) states: "In general, the soil water indices derived from the surface soil moisture observations and simulations agree well with the reference root-zone soil moisture."

Babaeian et al. (2018) states: "A close agreement was observed between the OPTRAM-SWDI and CMI drought indices for most selected sites. In conclusion, OPTRAM can estimate temporal soil moisture dynamics with reasonable accuracy for a range of climatic conditions (semi-arid to humid), soil types, and land covers, and can potentially be applied for agricultural drought monitoring."

Line 42. Please check English grammar.

**Authors:** Thank you very much. We are working to improve the English grammar.

Lines 46 to 49 please use references to support the drought definitions.

**Authors:** The following references already appear: Allaby, 2014; American Meteorological Society, 2004, 1997; Wilhite and Buchanan-Smith, 2005. If you consider it is necessary, we can include more references. At the beginning of the introduction:

"Drought is the principal climatic hazard to arid and semiarid Mediterranean grasslands and causes physical suffering, economic losses, and environmental degradation. Droughts are often divided into four major types: meteorological, agricultural, hydrological, and socioeconomic (Allaby, 2014; American Meteorological Society, 2004, 1997; Wilhite and Buchanan-Smith, 2005)."

---

## Author Comment (AC3)

**COMMENTS 2**

**Authors:** We are very grateful for the comments and suggestions to improve the study.

- In line 110, there is extra text. "Anexo 1- poner algo mas?" . You need to eliminate it

**Authors:** Done.

- In Figure 2, it is mentioned that it is used for the model in Equation 12 to estimate parameters. I believe it should say Equation 3.

**Authors:** Done.

- What is the motivation for reducing the value of w to 0-1 by replacing values of w above 1? If later a index (WCI that will range between 0 and 1) is used?

**Authors:** Although W and WCI range between 0-1, the meaning differs. WCI uses the expression:

$$WCI = \frac{W - W_{min}}{W_{max} - W_{min}}$$

where W is each value of the time series and $W_{min}$ and $W_{max}$ are, respectively, their multiyear minimum and maximum for every 10 days. The maximum and minimum values of the denominator reflect the best and worst conditions of surface soil moisture, respectively, and the difference between them somewhat reflects the condition of the local soil moisture.

- Where does Equation 3 come from (there is no reference, although it seems that it could be found in Sadaghi 2017 from Figure 2)? I think either a reference should be provided, or the foundation and interpretation of this index should be explained, or both.

**Authors:** We have improved the explanation of this index in the manuscript.

- In my opinion, the explanation of Equation 6 is unnecessary; I believe conditional probability should be well-known to everyone.

**Authors:** Done.

- In line 173, it is indicated how the phases are generated. To impose the limits of these phases, it seems like the median is used, although it is not explicitly stated. In my view, representing it with a Box-Cox diagram would allow a discussion of these points since the third quartile would suggest that phase 2 begins some weeks later, especially in Los Velez. Wouldn't it be clearer to exclusively represent a line connecting the medians of each ten-day period for greater clarity?

**Authors:** We added that the VCI median was used to delimit the phases and modified the figure 3 to include lines connecting the VCI and WCI medians.

- Figure 4 is not clear, especially regarding rainfall. It is difficult to distinguish those "abundant rains" in phase 3 (at least when comparing them with phase 2 where the extreme points are higher). There is also a contradiction stated in lines 188-189 regarding the explanation of Figure 4, as it indicates that there is more rain in phase 2.

**Authors:** Now it says: "Phases 3 and 4 in Bajo Aragón are very similar to Los Vélez with the same duration and time limits. The only difference is that the VCI maximum in Los Vélez is higher than in Bajo Aragón, probably, the vegetation of the area had a better condition to grow at Los Vélez in phase 2 (see Fig. 6A)".

- I suggest using more distinct colors to better distinguish the series in Figure 5 (for example, red and blue).

**Authors:** Done.

- In line 192, it is mentioned that there is a smoother profile of the ZVCI series; perhaps a cause explaining this should be suggested.

**Authors:** We have included the following explanation:

"This behaviour is revealing that vegetation response to environmental changes is slower than soil moisture response."

- Although providing probabilities in percentages may not be the most appropriate, it is understandable given the calculation method. However, I am more in favor of indicating the value between 0 and 1 when discussing probabilities.

**Authors:** In the methodology section, we have added an explanation about the calculation of the estimations of probabilities using percentual relative frequencies.

- Mentioning expected probabilities on standardized data could be considered. For example, if the indices followed normal distributions (which may not be the case), for ZWCI and ZVCI, the probabilities would be P(Index<-0.5)=0.309, P(Index<-0.7)=0.242, and P(Index<-0.1)=0.159.

**Authors:** We have added the following paragraph:

" In addition to this, probabilities of $Z_{WCI}$ and $Z_{WCI}$ for the different thresholds can be compared with expected probabilities on standardized data. If the indices, $Z_{WCI}$ and $Z_{WCI}$, followed normal distributions, the probabilities would be $P(Index < -0.5) = 0.309$, $P(Index < -0.7) = 0.242$, and $P(Index < -1.0) = 0.159$, which is not the case for the majority for the 10-day periods.

- The comments about the probability of an index being below the threshold of -0.7 being lower than being below -0.5 seem unnecessary. This is obvious: P(Z<-0.5)=P(Z<-0.7)+P(-0.7<=Z<-0.5) (the same for Z<-1). In any case, the greater the difference, the higher the probability that the index is between [-0.7, -0.5).

**Authors:** Thank you for your comment, you are right. However, thinking of potential readers, not specialists in statistics, we prefer to keep it.

- In line 215, why is there now a condition for the threshold of -0.3? I don't understand if there is justification or at least it is not explained.

**Authors:** The conditional probability we are obtaining is defined as:

- $P(Z_{VCI} < th \text{ at the period } i \mid Z_{WCI} < -0.3 \text{ at the same period } i)$:

The probability of an anomaly occurring in VCI at the period i (10-day) under the condition of an anomaly occurring in WCI ($Z_{WCI} < -0.3$) at the same period. "$th$" are the three different thresholds: -0.5, -0.7 and -1.0.

The reason to use the value -0.3 in $Z_{WCI}$ is to consider a great range of anomalies in the condition.

- In Figures 8, 9, 10, 11, P(VCI<-0.7) is stated. Obviously, this probability is always 0 because the VCI value is in the interval (0,1). It should be P(ZVCI<-0.7).

**Authors:** Thank you very much. You are right, there is a mistake in the legend of the graphs.

- In the paragraph between lines 224 and 230, there is no reference to the possibility that the correlation could be negative in the summer months (from late May to September). Additionally, the term "significantly" is used, but based on what criterion or test is it significant?

**Authors:** We have eliminated the term "significantly", the paragraph now says:

"Los Vélez shows high values in the lag-4 conditional probability with minor anomalies (-0.5 and -0.7), often reaching from 50% to 65% of probability from November to January (compared to an average of 20-30% of base probability). These probabilities decrease when threshold -1.0 is used, reaching a maximum of 50% in the middle of January."

- In lines 238 and 239, it is said, "as with Los Velez, the lag-4 conditional probabilities for the threshold -1.0 remain above the base probability." It seems incorrect to me; in Los Velez, these probabilities for the summer months were below.

**Authors:** Thank you very much. You are right, we have eliminated this paragraph.

- In line 241, it is said that the probability with a 4-period delay is higher than the base probability, but this also occurs for the probability without a delay.

**Authors:** You are right, there are some periods where lag-0 conditional probability is higher than lag-4 conditional probability, but in the discussion section we are focusing on the capability of predicting VCI anomalies, so we focused on lag-4 conditional probability.

- I wonder why, with two time series, a transfer function model is not carried out, identifying cross-correlations to determine the delay that best explains one series in terms of the other. It may have been done, but it is not indicated in the article. Why have you chosen 4 lag?

**Authors:** Previously, we did the cross-correlation to select optimal lag, but it is not included in the manuscript. If you consider it necessary, we can include it in an annexe.

---

## Author Comment (AC5)

**COMMENTS 1**

**Authors:** We are very grateful for the comments and suggestions to improve the study.

In this work the authors explore the feasibility of using water soil moisture (soil drought) as a warning index for vegetation drought. To this end they perform a study for two arid regions in Spain, for the period 2002-2019, with a 10-day temporal ad 250-500m spatial resolution. The conditional probability of the normalized Vegetation Condition Index (ZVCI) and Water Condition Index (ZWCI) with a 40-day lag is used to demonstrate that ZWCI can aid the prediction of vegetation anomalies ZVCI, particularly in the cooler months when vegetation growth is mainly driven by precipitation.

- The paper is clear and well written, it employs simple and sound methodology, and I believe that it represents an important contribution to the field of study. Apart from correcting a few typos e.g. "(Appendix 1- PONER ALGO MAS?)" on line 110, I would suggest to the authors to increase the figure labels as they are very hard to read.

**Authors:** We have corrected the mistake and replaced the figures the labels and legends are hard to read.

- Regarding the fact that both temperature and precipitation drive vegetation growth (as discussed along the paper), is there a way that temperature anomalies could also be taken into account explicitly in such a study?

**Authors:** We appreciate your suggestion. Indeed, we are carrying out other studies with precipitations. Studies with temperature could be the following research.